# Turning off the tap: Common domestic water conservation actions insufficient to alleviate drought in the United States of America

Diana M. Ruiz[1]◉*, Heather Tallis[2]◉, Bernie R. Tershy[1]‡, Donald A. Croll[1]‡

**1** Department of Ecology and Evolutionary Biology, University of California Santa Cruz, Santa Cruz, California, United States of America, **2** The Nature Conservancy, Santa Cruz, California, United States of America

◉ These authors contributed equally to this work.
‡ These authors also contributed equally to this work.
* dmadriga@ucsc.edu

**Data Availability Statement:** All US county water withdrawal and availability data files are available from the USGS database (https://waterdata.usgs.gov/nwis). All 2010 Census data files are available

## Abstract

Climate change is exacerbating drought and water stress in several global regions, including some parts of the United States. During times of drought in the U.S., municipal governments, private water suppliers and non-profits commonly deploy advocacy campaigns and incentive programs targeting reductions in residential water use through actions including: repairing leaks, shutting off taps, and installing new water-saving appliances. We asked whether these campaigns have the potential to alleviate water stress during drought at the county scale by estimating the potential impact of full adoption of such actions. In 2010, we show that the maximum potential use reductions from these residential actions may only alleviate water stress in 6% (174) of U.S. counties. The potential impact of domestic programs is limited by the relative dominance of agriculture water withdrawal, the primary water user in 50% of U.S. counties. While residential actions do achieve some water demand savings, they are not sufficient to alter water stress in the majority of the continental U.S. We recommend redirecting advocacy efforts and incentives to individual behaviors that can influence agricultural water use.

## Introduction

Drought intensity and frequency are increasing in some regions of the United States [1] resulting in increasing public attention to water conservation. For example, regionally, 30–40% of the western U.S. has experienced sustained drought in recent decades [2] and the concurrence of drought and heat waves has increased in duration and frequency across the US from 1990–2010 [3]. Over the same timeframe, human demand has increased in these areas leading to predictions that consumptive needs may not be met, such as along the Colorado River [4–7]. These stresses have prompted greater public interest in contributing to water conservation, with the United States ranking as the country with the highest proportional internet search for the terms "water savings" in 2018 [8]. Adults surveyed nationwide in 2013 indicated that 87%

from the United States Census Bureau database (https://www.census.gov/data.html).

**Funding:** The author(s) received no specific funding for this work.

**Competing interests:** The authors have declared that no competing interests exist.

were willing to conserve water to combat drought and over 35% of respondents believed the public should be the first to conserve water ahead of industry, cities, and agriculture [9].

Common public advocacy programs for water conservation promote individual-based strategies directed at reducing domestic water use [10, 11]. However, hydrologic and agronomic studies indicate that the agricultural sector, not the domestic sector, typically dominates water withdrawals [12, 13]. Within the agricultural sector, different food products and farming systems have widely varying water footprints while diet and product choices have the potential to exert a large influence on agricultural water demands [14–16]. Despite this knowledge, advocacy and incentive programs continue to focus on individual domestic behaviors and water conservation options on the apparent assumption that these activities have the potential to significantly contribute to water stress alleviation during drought. To date, no studies have evaluated whether these measures can reduce total water withdrawals sufficiently to alleviate water stress across the U.S. If currently promoted actions do not contribute sufficiently to alleviate water stress, it may be more effective and efficient to socialize and incentivize other individual actions that have greater impact. The most commonly promoted public water conservation actions include: repairing household leaks, taking shorter showers, closing faucets whenever possible, running dishwashers and washing machines only when full, and installing low-flow appliances [10, 17–27]. We examined whether full adoption of these actions in 2010 (a drought year) could lead to annual water savings sufficient to relieve water stress at the county scale across the continental U.S. Our intent is not to prescribe specific actions within a county or characterize detailed hydrologic processes and water stress conditions. Rather, we ask at a high level whether strongly-supported domestic efficiency practices are likely to alleviate water stress in the U.S.

## Materials and methods

We applied a simplified approach to estimate order of magnitude impacts of domestic efficiency methods on county scale water stress in the lower 48 states of the U.S. We determined which counties were water stressed in 2010 and calculated adjusted county level water use by assuming total household adoption of promoted domestic water conservation actions. Adjusted withdrawal calculations were used to determine which counties would change status from water stressed to non-water stressed under full adoption of common domestic efficiency practices in 2010.

### Water use sector

Sector-specific water use data were compiled from United State Geological Survey (USGS) records for 2010 [28], the most recent drought year with complete water monitoring data. Drought in the contiguous U.S. was more extensive between 2001–2010 than at any other time since the 1950s [29]. Total annual withdrawal rates (saline + fresh) for all US counties were binned into three sectors: industry, agriculture, and domestic. The industrial sector included water withdrawal used for industry, mining, and thermoelectric power. The agricultural sector captured water used for irrigation, livestock, and aquaculture. The domestic sector included water used for domestic and public supply. We calculated percent water withdrawal by sector and designated dominant water use sectors (accounting for 50% or more of total county withdrawals) for each county.

### Water stress

We defined water stress based on the ratio of water withdrawal to water availability for each county [30–33]. We considered any county with a water stress ratio at or above 0.4 to be water

stressed, adopting the commonly used thresholds in the water drought and scarcity literature [3–6]. Ideally, water stress values would be calculated at a high temporal resolution to capture variations in important hydrologic drivers (rainfall, baseflow, groundwater recharge rates, evapotranspiration rates, etc.) and demand drivers (temperature, growing seasons, competing demands, water prices, etc.). However, monthly or daily water withdrawal (demand) data are not publicly available at the county level in the U.S. Given the paucity of temporally variable demand data, we opted for a simple water stress calculation rather than complex modeling exercises. As our intent was to understand whether domestic efficiency changes are of the order of magnitude to alleviate water stress, we consider this a sufficient order of magnitude approximation approach.

Water withdrawal rates were defined as total annual saline and fresh water withdrawals [28]. Water availability was calculated monthly from USGS data [20] as the difference between total monthly evapotranspiration (L) and total monthly precipitation (L). Monthly county water availability values were then summed across the year to generate an annual water availability metric for each county. Reported annual water withdrawal values were then related to these annual availability values to estimate a water stress ratio per county as described above.

This simplified approach does not represent important temporal relationships between rainfall, extraction, return flows, or groundwater, and only considers locally available surface water supplies, overlooking major inter-basin water transfers. In addition, analyses were bounded by political governance boundaries (counties) rather than watersheds. Such simplified calculations have been used in other contexts to explore relative water stress conditions [34–36], and were deemed sufficient here to approximate the significance of domestic efficiency approaches and to provide an estimate in the absence of more temporally resolved withdrawal data.

## Conservation-adjusted withdrawals

We calculated the potential water savings per household from full adoption of twelve commonly recommended and reliably measured water conservation actions (Table 1). This criteria led to the exclusion of actions not found to be regularly recommended by water governing or advocacy groups or for which consistent savings could not be calculated, for example, limiting toilet flushing, washing cars on lawns, or changing household water pressure. While our study

**Table 1. Standardized potential monthly water savings per household from commonly promoted domestic water conservation actions.**

| Water Conservation Action and Reference | Water Savings (L household$^{-1}$ month$^{-1}$) |
|---|---|
| Repair household leaks [17, 38] | 3155 |
| Shorten shower time from 8 minutes to 5 minutes [18] | 2930 |
| Close faucet when brushing teeth [19] | 2344 |
| Install WaterSense labeled low flow toilets [20] | 1322 |
| Install WaterSense showerheads [22] | 726 |
| Install ENERGYSTAR clothes washer [26] | 431 |
| Only run full loads in dishwasher [21] | 399 |
| Close faucet when shaving [21] | 333 |
| Install WaterSense irrigation controller [23] | 324 |
| Install ENERGYSTAR dishwasher [24] | 316 |
| Install WaterSense bathroom faucet and aerators [25] | 158 |
| Close faucet while hand washing or rinsing dishes [27] | 46 |
| Total Maximum Potential Water Savings | 14,213 |

does not incorporate all possible domestic water conservation actions, it represents those most commonly recommended by government and activist groups. Assuming full adoption of these actions provides an upper bound estimate of the largest impact that common water efficiency campaigns could have on domestic water use. We gathered estimates of potential savings per U.S. household for each action and applied the most conservative estimates for all calculations. When data permitted, projected savings from new installations were adjusted to exclude households already fitted with water efficient appliances (S1 Table). Likewise, actions influenced by age or sex were also adjusted to reflect the county-level demographics of the population assumed to adopt efficiency actions. Annual single household savings was based on U.S. average household occupancy in 2010 [37]. Annual single household savings estimates were multiplied by the number of households in each county [37] to estimate county-level potential annual water savings. We then determined the impact of these domestic actions on water stress by subtracting our estimated county-level potential annual water savings from reported total annual withdrawal rates and recalculating water stress for all counties as above. We could not account for differences in savings that might result from variance in household water pressure because we could not identify a data source capturing this variance and installation of pressure regulating devices at the county level.

Water savings for interventions estimates were primarily drawn from water governing agencies reports such as the US Ecological Protection Agency and local governments. Further research is needed to characterize county specific current water use practices and potential gains. Rebate programs provide important information on water efficient replacements; however miss households that do not participate in the programs. Here estimates on potential water savings assume that national water use practices averages can be applied across counties. For the nationwide view of this study, these assumptions are permissible.

## Results

In 2010, the majority (92%) of the 3,109 US counties in the lower 48 states were water stressed based on our basic water stress indicator (Fig 1). As 2010 was a drought year, this high rate of county-level water stress was expected. Estimated reductions in domestic water withdrawals from full household adoption of commonly supported water saving measures would release 174 counties (6% of all continental counties) from water stress (Fig 2). Domestic water use dominated in these mostly rural, eastern and mid western counties.

Including the potential impact of domestic water saving actions, 86% of counties remained water stressed. This shortfall is likely explained by the dominance of the agricultural sector withdrawals in most counties (Fig 3). Of the 2,242 counties with a dominant sector, agriculture was dominant in approximately twice as many counties (1,103 counties) as either the domestic (570 counties) or industrial (569 counties) sectors.

## Discussion

Given these findings, the potential impact of domestic savings may be inflated relative to potential savings in other sectors. The national dominance of agricultural withdrawals suggests that water stress will not be alleviated in most counties unless agricultural water efficiencies are achieved. Further emphasizing the need to address agricultural water withdrawals, this sector's water use dominates in regions projected to experience the greatest future increases in drought [1, 39, 40].

There is a substantial literature describing measures for individuals to influence agricultural sector water withdrawals. For example, individuals may tailor their diet to reduce their water footprint. Additionally, consumers could pressure suppliers to reveal information about how

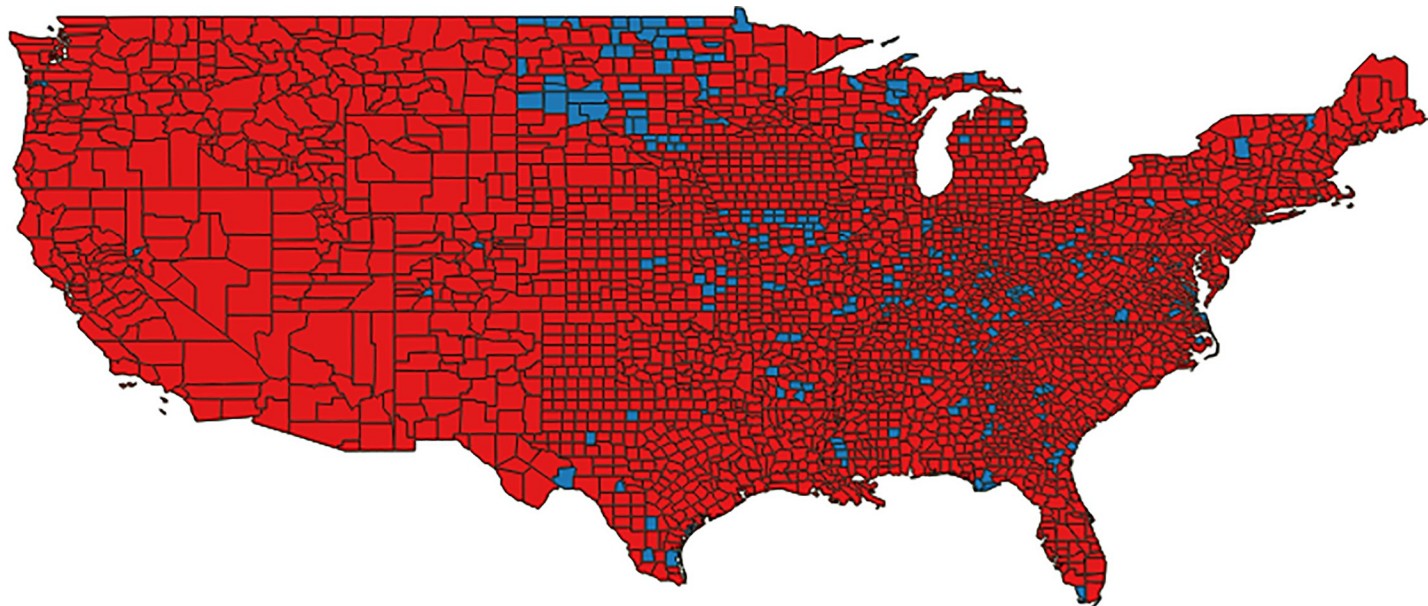

**Fig 1. Water stress status of U.S. counties in 2010, a drought year.** The majority of counties (2,853) in the continental United States were water stressed (red), while only 256 counties were free from water stress (blue).

well matched crops are to growing region conditions. For example, cotton, paddy rice, alfalfa and other high water use crops grown in wetter regions are less likely to stress local water supplies than the same crops grown in drier regions. Several major companies such as Wal-Mart and Kellogg's have started to improve their supply chain tracking and could begin to provide this kind of information if buyers demanded it. Consumers could also reduce their portion of food waste, and encourage farmers to reduce waste that happens in harvesting and processing,

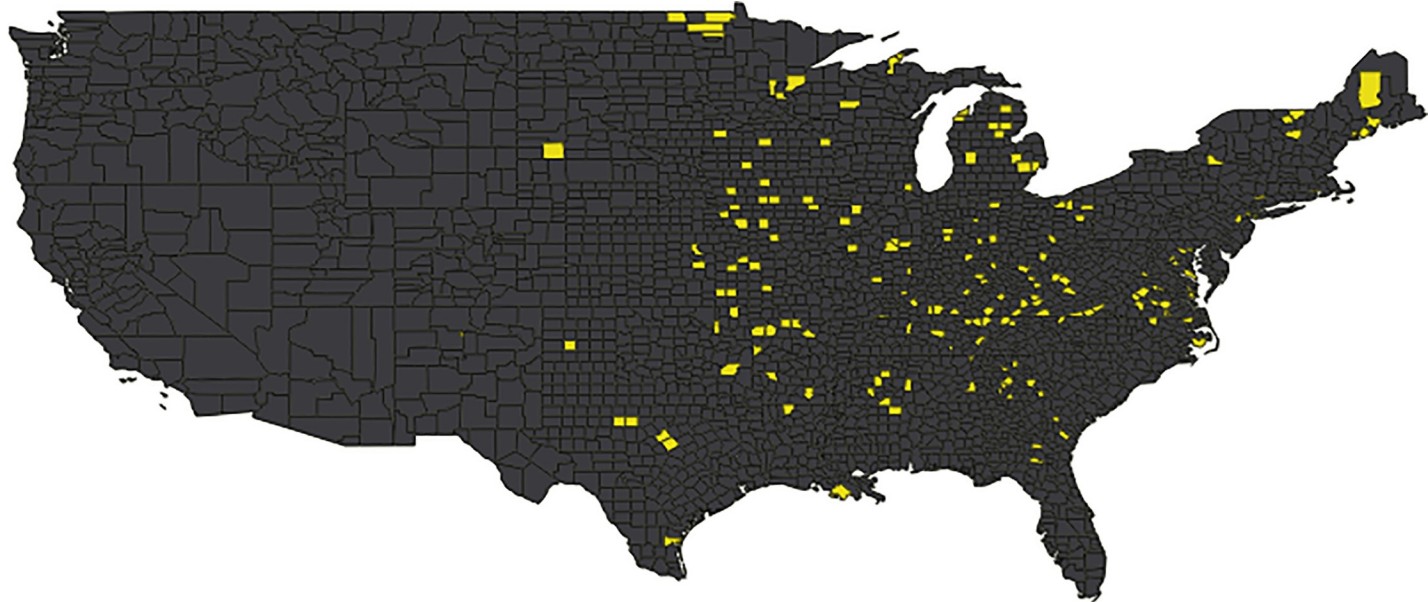

**Fig 2. Counties with potential water stress relief from promoted domestic actions.** Only 174 (6%) of all counties have the potential to be relieved from water stress (yellow) by full adoption of conventionally supported household water conservation actions.

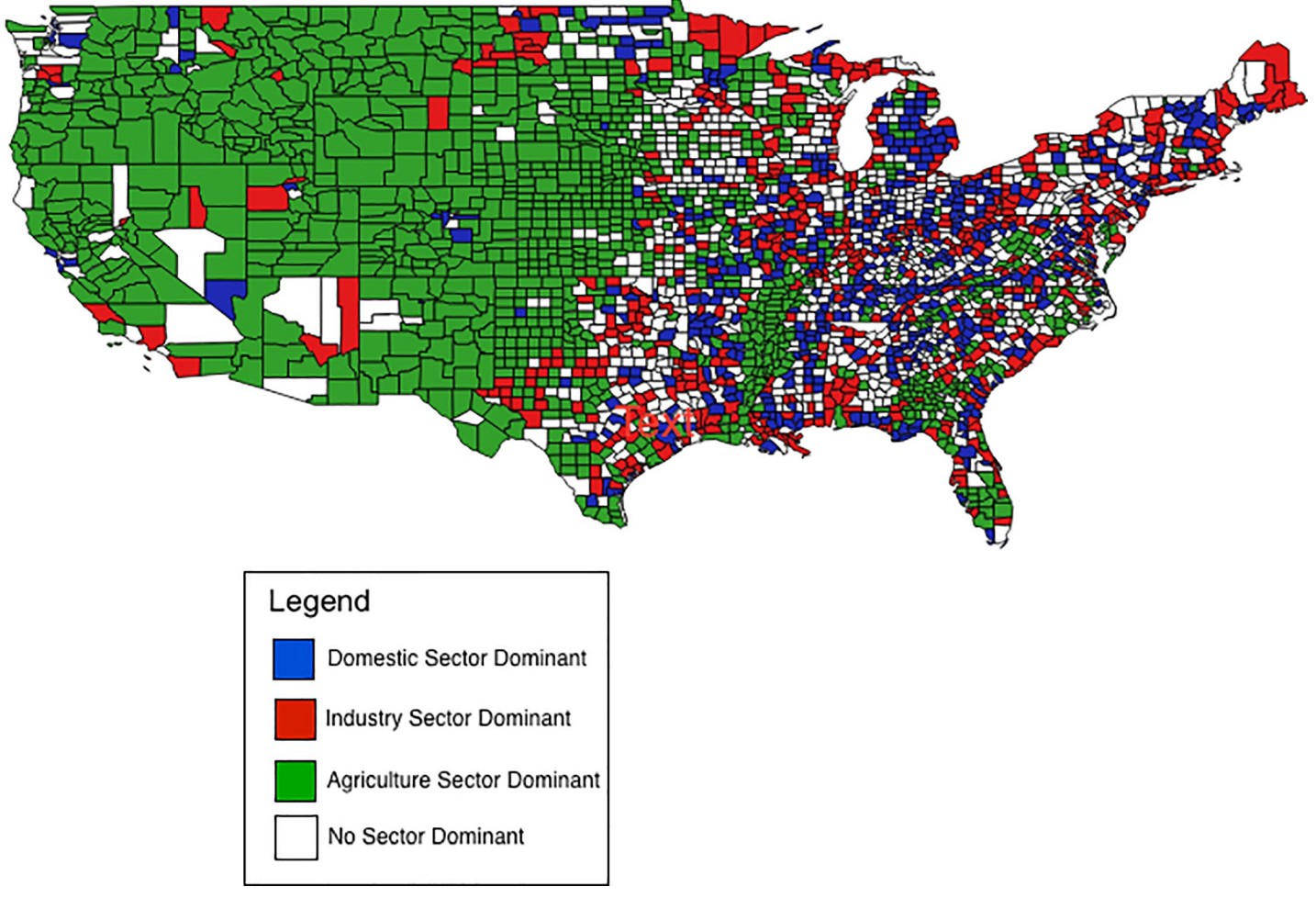

**Fig 3. 2010 County level dominant water withdrawal sectors.** Blue indicates >50% water withdrawal attributed to the domestic sector. Red indicates >50% of water withdrawal attributed to the industrial sector. Green indicates >50% of water withdrawal attributed to the agricultural sector.

minimizing the amount of water 'lost' through the production and disposal of uneaten food [41–43].

However, these potential water conservation measures are seldom recommended in water-saving campaigns, while the domestic actions evaluated here dominate. Our analyses suggest that domestic water savings advocacy and incentive programs will fail the majority of the time in the United States because domestic water use is not the dominant driver of water stress, and available household savings measures are not sufficient to transition the majority of counties out of water stress during a drought. Promoting individual water conservation actions targeted at reducing agricultural withdrawal have a much higher potential to significantly improve water savings and promote longer term water security.

## Supporting information

**S1 Table. Commonly recommended water conservation actions in the U.S. and associated estimated household savings in 2010.**
(PDF)

## Author Contributions

**Conceptualization:** Diana M. Ruiz, Heather Tallis, Bernie R. Tershy, Donald A. Croll.

**Methodology:** Diana M. Ruiz.

**Project administration:** Heather Tallis.

**Supervision:** Heather Tallis.

**Validation:** Heather Tallis.

**Writing – original draft:** Diana M. Ruiz.

**Writing – review & editing:** Diana M. Ruiz, Heather Tallis, Bernie R. Tershy, Donald A. Croll.

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
