## [Decision Letter · Decision Letter 0]

30 Dec 2019

PONE-D-19-28748

Turning off the tap: Common domestic water conservation actions insufficient to alleviate drought in the United States of America

PLOS ONE

Dear Mrs Ruiz,

Thank you for submitting your manuscript to PLOS ONE. After careful consideration, we feel that it has merit but does not fully meet PLOS ONE’s publication criteria as it currently stands. Therefore, we invite you to submit a revised version of the manuscript that addresses the points raised during the review process.

**Please add some your Options for the Future: Balancing Water Demand and Water Resources at domestic level **

We would appreciate receiving your revised manuscript by Feb 13 2020 11:59PM. To enhance the reproducibility of your results, we recommend that if applicable you deposit your laboratory protocols in protocols.io, where a protocol can be assigned its own identifier (DOI) such that it can be cited independently in the future. For instructions see: http://journals.plos.org/plosone/s/submission-guidelines#loc-laboratory-protocols

We look forward to receiving your revised manuscript.

Kind regards,

Shahid Farooq

Academic Editor

PLOS ONE

Journal Requirements:

2. Please amend your list of authors on the manuscript to ensure that each author is linked to an affiliation. Authors’ affiliations should reflect the institution where the work was done (if authors moved subsequently, you can also list the new affiliation stating “current affiliation:….” as necessary).

Reviewers' comments:

Reviewer's Responses to Questions

**Comments to the Author**

1. Is the manuscript technically sound, and do the data support the conclusions?

Reviewer #1: Yes

2. Has the statistical analysis been performed appropriately and rigorously? 

Reviewer #1: Yes

3. Have the authors made all data underlying the findings in their manuscript fully available?

Reviewer #1: Yes

4. Is the manuscript presented in an intelligible fashion and written in standard English?

Reviewer #1: Yes

5. Review Comments to the Author

Reviewer #1: Reviewer comments

Turning off the tap: Common domestic water conservation actions insufficient to alleviate drought in the United States of America

PONE-D-19-28748

Please add some your Options for the Future: Balancing Water Demand and Water Resources at domestic level

Please estimated water savings for a house with low pressure in county then compare to a house with high pressure, your data will be more reliable for future estimation if you please add some data regarding important voluntary domestic water conservation measures include the following:

• Limiting toilet flushing.

• Adopting water-saving plumbing fixtures, such as toilets and shower heads.

• Adopting water-efficient appliances (notably washing machines).

• Limiting outdoor uses of water, as by watering lawns and gardens during the evening and early morning, and washing cars on lawns and without using a hose.

6. PLOS authors have the option to publish the peer review history of their article (what does this mean?). If published, this will include your full peer review and any attached files.

Reviewer #1: No

---

## [Author Response · Author response to Decision Letter 0]

9 Feb 2020

Reviewer comment: 

“Please add some your Options for the Future: Balancing Water Demand and Water Resources at domestic level

 Please estimated water savings for a house with low pressure in county then compare to a house with high pressure,”

Response: 

We are unclear as to the reviewer’s use of the term “pressure” here, and so respond on the basis of two possible interpretations. The reviewer may be using pressure to refer to water pressure entering the household through pipes from the municipality or well. In this case, it would be expected that a household with high water pressure would use more water than a low water pressure household, and so water saving devices may have larger benefit when installed in high water pressure households. To analyze this variation, we would need to know average water pressure in each county, and make an assumption about how many houses already have pressure regulating devices installed. Water pressure can vary dramatically even within the service area of one municipal water provider (based on distance from source, elevation, pipe diameter, etc). We were unable to identify an available data source that would allow us to robustly estimate impact of water pressure on adoption of other water saving devices. We have added text to the methods section clarifying this limitation, lines 152-155.

 A second interpretation of “pressure” in the reviewer’s comment could relate to household water demand, wherein high pressure would represent households with high water demand. Our methods relied on total domestic demand reported at the county level. We are not aware of data avialable at the individual household level and so are unable to make household level demand comparisons. 

Reviewer comment:

“your data will be more reliable for future estimation if you please add some data regarding important voluntary domestic water conservation measures include the following:

• Limiting toilet flushing.

• Adopting water-saving plumbing fixtures, such as toilets and shower heads.

• Adopting water-efficient appliances (notably washing machines).

• Limiting outdoor uses of water, as by watering lawns and gardens during the evening and early morning, and washing cars on lawns without using a hose.”

Response: 

Our chosen water saving interventions include all elements suggested here, except limiting toilet flushing. Our original manuscript includes water saving plumbing fixtures as specified for toilets (in Table 1, row 4 “Install WaterSense labeled low flow toilets”), showerheads (Table 1, row 5 “Install WaterSense showerheads”) and washing machines (Table 1, row 6 “Install ENERGYSTAR clothes washer”). We also include the commonly recommended action to limit outdoor water use as the installation of a WaterSense irrigation controller (Table 1, row 9). WaterSense irrigation controllers earn the WaterSense label by meeting EPA standards for avoiding overwatering. These irrigation controls assess actual local weather conditions to determine watering needs rather than preset timers. We feel this adequately addresses the action of efficient outdoor watering in the most reliably measured manner. All of the mentioned actions and the derivation of their estimated savings are also found in Supplemental Table 1.

 As washing cars on lawns and limiting toilet flushing were not found to be a commonly recommended action and difficult to reliably estimate potential savings, we have not included it in our analysis. However, we have added text to note such exclusions in our methods, lines 133-140.

---

## [Editor Report · Decision Letter 1]

14 Feb 2020

Turning off the tap: Common domestic water conservation actions insufficient to alleviate drought in the United States of America

PONE-D-19-28748R1

Dear Dr. Ruiz,

We are pleased to inform you that your manuscript has been judged scientifically suitable for publication and will be formally accepted for publication once it complies with all outstanding technical requirements.

With kind regards,

Shahid Farooq

Academic Editor

PLOS ONE

Additional Editor Comments (optional):

I have evaluated the revised manuscript. The authors have addressed the minor points raised during review process. Thus, the manuscript can be accepted in its current form.
---

## [Editor Report · Acceptance letter]

19 Feb 2020

PONE-D-19-28748R1 

Turning off the tap: Common domestic water conservation actions insufficient to alleviate drought in the United States of America 

Dear Dr. Ruiz:

I am pleased to inform you that your manuscript has been deemed suitable for publication in PLOS ONE. Congratulations! Your manuscript is now with our production department. 

With kind regards,

on behalf of

Dr. Shahid Farooq 

Academic Editor

PLOS ONE